# Fast Scalable Architecture of a Near-ML Detector for a MIMO-QSM Receiver

**Ismael Lopez [1], L. Pizano-Escalante [2], Joaquin Cortez [1],***  **and O. Longoria-Gandara [2] and Armando Garcia [1]**

[1]   Electronics and Electrical Engineering Department, Instituto Tecnologico de Sonora,
     Ciudad Obregon 85000, Mexico; ilopez@firstpassglobal.com (I.L.); armando.garcia@itson.edu.mx (A.G.)
[2]   Department of Electronics, Systems and Computer Science, Instituto Tecnologico y de Estudios Superiores
     de Occidente, Tlaquepaque 45604, Mexico; luispizano@iteso.mx (L.P.-E.); olongoria@iteso.mx (O.L.-G.)
*   Correspondence: joaquin.cortez@itson.edu.mx

**Abstract:** This paper presents a proposal for an architecture in FPGA for the implementation of a low complexity near maximum likelihood (Near-ML) detection algorithm for a multiple input-multiple output (MIMO) quadrature spatial modulation (QSM) transmission system. The proposed low complexity detection algorithm is based on a tree search and a spherical detection strategy. Our proposal was verified in the context of a MIMO receiver. The effects of the finite length arithmetic and limited precision were evaluated in terms of their impact on the receiver bit error rate (BER). We defined the minimum fixed point word size required not to impact performance adversely for $n_T$ transmit antennas and $n_R$ receive antennas. The results showed that the proposal performed very near to optimal with the advantage of a meaningful reduction in the complexity of the receiver. The performance analysis of the proposed detector of the MIMO receiver under these conditions showed a strong robustness on the numerical precision, which allowed having a receiver performance very close to that obtained with floating point arithmetic in terms of BER; therefore, we believe this architecture can be an attractive candidate for its implementation in current communications standards.

**Keywords:** FPGA; QSM; MIMO; Near-ML detection

## 1. Introduction

A wireless communications system employing multiple antennas achieves a better performance over the wireless channel in terms of capacity and bit error rate (BER) [1] by spreading the transmitted information over space and time in a pattern specified by a space-time block code [2]. Classic examples include V-BLAST, which maximizes the spatial multiplexing gain [3], and the Alamouti code [4], which maximizes the diversity gain. A wide variety of space-time coding techniques has been proposed in the past two decades, each achieving a specific combination of rate and diversity gains and whose decoding requires a certain computational complexity.

Recently, the spatial dimension of MIMO systems was used to modulate the transmitted signal instead of using it for diversity or multiplexing. This technique is known as spatial modulation (SM) [5–7]. SM considers the complete array of transmit (Tx) antennas as a spatial constellation, where each Tx antenna in this array represents a point in the spatial constellation. In SM, only one Tx antenna is activated at a time, while the other Tx antennas stay off the air during one symbol transmission. Ideally, there is a different and unique channel between each Tx antenna and a receive (Rx) antenna. Therefore, the receiver can determine which Tx antenna has been utilized

in the transmission. In this way, spatially modulated bits add to the received quadrature amplitude modulation (QAM) symbol [8].

Quadrature spatial modulation (QSM) is an SM based transmission technique that uses the quadrature (Q) and the in-phase (I) QAM components to independently modulate a spatial constellation. As a result, QSM has better spectral efficiency (SE) since it transmits twice the number of bits in the spatial domain [9–11] with the aim of further improving the performance of SM systems in terms of bit error rate.

The complexity in the receiver is an important factor to take into account for hardware implementations [12], mainly in MIMO and massive MIMO systems where the number of antennas used increases significantly the complexity of the receivers. Some previous works have analyzed this problem for basic QSM transmission schemes. For example, low complexity QSM detectors that consider compressive sensing [13,14], sphere decoding [15], minimum mean squared error (MMSE) [16], equivalent maximum likelihood [17], and zero forcing precoding [18] based low complexity detectors have been recently proposed. However, to the best of our knowledge, practical implementations of hardware architectures for the detection process in MIMO-QSM systems have not been discussed.

Against this background, this research proposes an architecture in FPGA to implement a low complexity near maximum likelihood (Near-ML) detector for a MIMO-QSM system. The proposed low complexity detection algorithm is compared with the ML criterion in terms of BER performance and complexity. Results show that the proposed low complexity Near-ML algorithm performs very near to the optimal detector while achieving a complexity reduction of up to 88% for the analyzed cases.

The contribution of the paper is three-fold:

- A modified low complexity Near-ML detection algorithm based in sphere detectors for the MIMO-QSM system is proposed.
- A new reconfigurable architecture to implement in FPGA the Near-ML detection algorithm is proposed.
- The proposed architecture for the MIMO-QSM receiver can operate for different combinations in transmitter and receiver antennas, including all different sizes of modulation schemes *M*-QAM, which makes it attractive to be used in the new wireless communication standards.

The remainder of the paper is organized as follows: A description of the QSM general system model with a brief explanation of the QSM implementation, the channel effects at the QSM receiver, and the considerations for an ML detection are presented in Section 2; Section 3 describes the low complexity detection algorithm implemented at the MIMO-QSM receiver; Section 4 show a full description of the proposed architecture for the MIMO-QSM receiver; Section 5 shows the results of BER performance and the proposed architecture to implement the receiver for the MIMO-QSM system. Finally, conclusions are summarized in Section 6.

*Notation*: Uppercase boldface letters denote matrices, whereas lowercase boldface letters denote vectors. The transpose, Hermitian transpose, and Frobenius norm of $\mathbf{A}$ are denoted by $\mathbf{A}^T$, $\mathbf{A}^H$, and $\|\mathbf{A}\|_F^2$, respectively. Finally, $\mathcal{CN}(\mu, \sigma^2)$ is used to represent the circularly symmetric complex Gaussian distribution with mean $\mu$ and variance $\sigma^2$.

## 2. MIMO-QSM Transmission System

The system model of the MIMO-QSM transmission scheme is presented in Figure 1. We considered a transmitter with $n_T$ transmit antennas and a receiver with $n_R$ receiving antennas. Thus, the end-to-end configuration can be considered as a $n_R \times n_T$ MIMO transmission system. We assumed a rich scattering, Rayleigh wireless channel with flat and slow fading, where the channel between transmitter antenna $j$ and receiver antenna $i$ can be modeled as a complex Gaussian gain $h_{ij} \sim \mathcal{C}(0, 1)$ of zero mean and variance 0.5 per dimension. This gain remains constant for several symbol intervals, after which it changes to a new independent channel realization. The system considered here can transmit $m_{QSM} = \log_2(M) + 2\log_2(n_T)$ bits in each time slot, where $M$ is the size of the $M$-ary quadrature

amplitude modulation (QAM) constellation $\mathcal{S} = \{s_1, s_2, \cdots, s_M\}$ and $n_T$ is the size of the QSM transmission vector. Thus, the general MIMO-QSM communication system can be mathematically modeled as:

$$
\begin{bmatrix} y_1 \\ \vdots \\ y_{n_R} \end{bmatrix} = \sqrt{\gamma} \begin{bmatrix} h_{1,1} & \cdots & h_{1,n_T} \\ \vdots & \ddots & \vdots \\ h_{n_R,1} & \cdots & h_{n_R,n_T} \end{bmatrix} \begin{bmatrix} x_1 \\ \vdots \\ x_{n_T} \end{bmatrix} + \begin{bmatrix} n_1 \\ \vdots \\ n_{n_R} \end{bmatrix},
\tag{1}
$$

which can be expressed equivalently as:

$$
\mathbf{y} = \sqrt{\gamma} \mathbf{H} \mathbf{x} + \mathbf{n},
\tag{2}
$$

where $\mathbf{x} \in \mathbb{C}^{n_T \times 1}$ is the overall transmission vector and $\mathbf{y} \in \mathbb{C}^{n_R \times 1}$ is the received vector. $\mathbf{H} \in \mathbb{C}^{n_R \times n_T}$ is the channel matrix; $\gamma$ is the signal-to-noise ratio (SNR) at each antenna; and $\mathbf{n} \in \mathbb{C}^{n_R \times 1}$ represents the AWGN vector. The generated noise samples are independent and identically distributed (i.i.d.) with $\mathcal{CN}(0, \sigma^2)$.

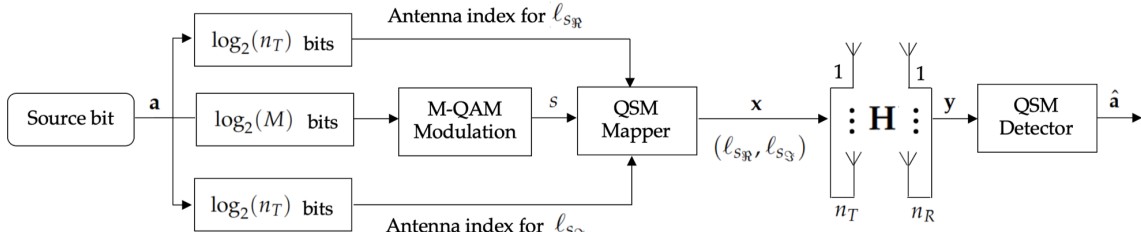

**Figure 1.** The MIMO-QSM system model.

Further assumptions were that all the antennas transmit information symbols from the same *M*-QAM constellation, the receiver has perfect channel state information (CSI), and the receiver is perfectly synchronized to the transmitter.

### 2.1. QSM Modulation

In order to generate the QSM signals $\mathbf{x}$, the input sequence of bits $\mathbf{a}$ was split into three flows. One flow was utilized to modulate an *M*-QAM signal, and the other two flows (spatial bits) were used to modulate the position in the output vector $\mathbf{x}$. For an input bit sequence of length $m_{QSM}$ bits, the first $\log_2(M)$ bits modulated an *M*-QAM symbol. The remaining $2\log_2(n_T)$ bits were split into two flows of $\log_2(n_T)$ spatial bits. These spatial bits modulated the position in the output vector $\mathbf{x}$ as follows: the real part of the QAM symbol was assigned to one specific position in the output vector of length $n_T$. The imaginary part of the QAM symbol was assigned to another one or even the same position antenna in the output vector $\mathbf{x}$. Finally, these two SM signals were combined to obtain the QSM output vector [9].

Table 1 shows a mapping rule example for QSM using four-QAM and $n_T = 2$. The first column shows the input bit sequence of length $m_{QSM}$, where the first two bits modulate a four-QAM symbol and the remaining two bits modulate the position of the non-zero entries in the output vector $\mathbf{x}$ as follows: the real part of the QAM symbol was assigned to one Tx antenna out of $n_T$ available in order to modulate $\log_2(n_T) = 1$ bit, whereas the imaginary part of the QAM symbol was assigned to another or even the same position to modulate $\log_2(n_T) = 1$ bit to define the $\ell_\Re$ and $\ell_\Im$ index transmitter antenna, respectively, as shown in the third column of Table 1.

**Table 1.** Example of the QSM mapping rule with $n_T = 2$.

| Input | QAM Symbol | Position Antenna | Output Vector |
|:-----:|:----------:|:----------------:|:-------------:|
| a | s | $(\ell_\Re, \ell_\Im)$ | $x_i$ |
| 0000 | $1 + j$ | (1, 1) | $1 + j, 0$ |
| 0001 | $1 + j$ | (1, 2) | $1, +j$ |
| 0010 | $1 + j$ | (2, 1) | $j, +1$ |
| 0011 | $1 + j$ | (2, 2) | $0, 1 + j$ |
| 0100 | $-1 + j$ | (1, 1) | $-1 + j, 0$ |
| 0101 | $-1 + j$ | (1, 2) | $-1, +j$ |
| 0110 | $-1 + j$ | (2, 1) | $j, -1$ |
| 0111 | $-1 + j$ | (2, 2) | $0, -1 + j$ |
| 1000 | $-1 - j$ | (1, 1) | $-1 - j, 0$ |
| 1001 | $-1 - j$ | (1, 2) | $-1, -j$ |
| 1010 | $-1 - j$ | (2, 1) | $-j, -1$ |
| 1011 | $-1 - j$ | (2, 2) | $0, -1 - j$ |
| 1100 | $1 - j$ | (1, 1) | $1 - j, 0$ |
| 1101 | $1 - j$ | (1, 2) | $1, -j$ |
| 1110 | $1 - j$ | (2, 1) | $-j, +1$ |
| 1111 | $1 - j$ | (2, 2) | $0, 1 - j$ |

*2.2. ML Detection*

The MIMO transmitted signal is:

$$\mathbf{r} = \sqrt{\gamma}\mathbf{H}\mathbf{x} \tag{3}$$

where the transmitted symbol $\mathbf{x}$ can be split into two real valued signals, $s_\Re$ and $s_\Im$, according to the QSM mapping rule described in Table 1. Since the QAM symbol $s$ is expanded by the matrix $\mathbf{H} \in \mathbb{C}^{n_R \times n_T}$, (3) can be expressed as:

$$\mathbf{r} = \mathbf{h}^{(\ell_\Re)} \cdot s_\Re + j\mathbf{h}^{(\ell_\Im)} \cdot s_\Im, \tag{4}$$

where $\mathbf{h}^{(\ell_\Re)}$ and $\mathbf{h}^{(\ell_\Im)}$ denote the $\ell_\Re^{\text{th}}$ and $\ell_\Im^{\text{th}}$ columns of $\mathbf{H}$, respectively, with $\ell_\Re, \ell_\Im \in \{1, 2, \cdots, n_T\}$, and the vectors $\mathbf{s}_\Re = \Re\{\mathcal{S}\}$ and $\mathbf{s}_\Im = \Im\{\mathcal{S}\}$ of dimension $q_\Re$ and $q_\Im$, respectively, represent the different real and imaginary parts of the symbols belonging to the $M$-QAM constellation $\mathcal{S}$; finally, the symbol $\cdot$ represents the product among the $\ell_\Re^{\text{th}}$ and $\ell_\Im^{\text{th}}$ columns of $\mathbf{H}$ and each element of the vectors $\mathbf{s}_\Re$ and $\mathbf{s}_\Im$.

Assuming that the receiver had perfect channel state information, the ML estimation compared the distance between the received signal and all possible received signals. The ML criterion is defined as:

$$\hat{\mathbf{x}} = \underset{\mathbf{x} \in \mathcal{X}}{\arg\min} \parallel \mathbf{y} - \sqrt{\gamma}\mathbf{H}\mathbf{x} \parallel_F^2, \tag{5}$$

where $\parallel \parallel_F^2$ is the Euclidean distance among two vectors and $\mathcal{X} \in \mathbb{C}^{n_T \times 2^{m_{QSM}}}$ is the full spatial modulation QSM used in the transmitter; therefore, the ML detector jointly estimated the two possible active Tx antenna indices, $\hat{\ell}_\Re$ and $\hat{\ell}_\Im$, and the corresponding real valued signals $\hat{s}_\Re$ and $\hat{s}_\Im$. Therefore, (5) can be written as:

$$\left[\hat{\ell}_\Re, \hat{\ell}_\Im, \hat{s}_\Re, \hat{s}_\Im\right] = \underset{\ell_\Re, \ell_\Im, s_\Re, s_\Im}{\arg\min} \parallel \mathbf{y} - \mathbf{h}^{(\ell_\Re)} \cdot s_\Re - j\mathbf{h}^{(\ell_\Im)} \cdot s_\Im \parallel_F^2 \tag{6}$$

## 3. Low Complexity Detection Algorithm

In recent works, some low complexity algorithms have been presented for MIMO, SM, and QSM signal detection [10,13,19]. In the works presented by [14,16], optimization algorithms and trigonometric functions were required in the receiver to detect the most likely antenna combinations. After the antenna indices were detected, a reduced ML detector was utilized to identify the transmitted symbols in the MIMO-QSM system. However, these schemes demand many hardware implementation resources. Other detection techniques for MIMO-SM-QSM were reported in [15,19,20]. These detectors were based on tree search and spherical detection. These algorithms had an excellent BER performance, and their detection complexity in terms of flops were relatively simple for hardware implementation in FPGA.

In this section, a modified low complexity Near-ML detector for MIMO-QSM signals is presented. The proposed detector is based on a tree search and a spherical algorithm [20,21].

The ML solution to (6) may be expressed as a tree search: each branch in this tree was assigned a distance metric where the symbols with the smallest overall distance were selected as possible optimum solutions [22] in each level of the tree. To carry out this process, an adaptive *M*-algorithm based on a breadth first sorted tree search was used. The proposed algorithm reduced the search complexity by storing only, at maximum, the best *L* branches at a time [23] in each level of the tree. Henceforth, a small *L* resulted in low complexity and relatively sub-optimal performance. As *L* increased, the complexity of the detector in terms of flops also increased, and the performance of the algorithm approached the ML solution.

The decision metrics $\mathbf{d}_1$ and $d_T$, required in the Near-ML proposal detector for the MIMO-QSM scheme can be established as follows:

$$\mathbf{d}_1 = \| \mathbf{y} - \mathbf{h}^{(\hat{\ell}_\Re)} \cdot \hat{\mathbf{s}}_\Re \|_F^2, \tag{7}$$

$$d_T = \| \mathbf{y} - \mathbf{h}^{(\hat{\ell}_\Re)} \cdot \hat{\mathbf{s}}_\Re^{(l)} - \mathbf{h}^{(\hat{\ell}_\Im)} \cdot \hat{\mathbf{s}}_\Im^{(m)} \|_F^2 . \tag{8}$$

We denote the $l^{\text{th}}$ and $m^{\text{th}}$ element in the vectors $\mathbf{s}_\Re$ and $\mathbf{s}_\Im$ by $\mathbf{s}_\Re^{(l)}$ and $\mathbf{s}_\Im^{(m)}$, respectively. The goal of the decoder is to find the optimum solution to the ML criterion in (6), using the distances calculated in (7) and (8). For the case of the distance $\mathbf{d}_1$, it is a vector of distances calculated for each valid combination between the $\hat{\ell}_\Re^{\text{th}}$ transmitter antenna $\mathbf{h}^{(\hat{\ell}_\Re)}$ and each element of the vector $\mathbf{s}_\Re$. The decoding procedure was split into two parts. First, a pre-ordering with $\mathbf{s}_\Re$ was carried out; specifically, the distance of (7) was calculated, and symbols were re-ordered in ascending order. In this way, a set of $N_\Re = q_\Re n_T$ tuples was obtained, each tuple $\left( \hat{\ell}_\Re^{(l)}, \hat{s}_\Re^{(l)} \right)$ being formed of a combination of the Tx antenna index $\hat{\ell}_\Re \in \{1, \cdots, n_T\}$ and the Tx symbol $\hat{s}_\Re^{(l)}$. This part is summarized in Algorithm 1.

The second part corresponds to an optimized detector based on the detector for SM signals published in [21]. Since the first part of the Near-ML algorithm estimated the real part of the transmitted QSM symbol, $\left( \hat{\ell}_\Re^{(l)}, \hat{s}_\Re^{(l)} \right)$, the second part considered that only one Tx antenna was active, which corresponded to the imaginary part of the QSM transmitted symbol $\left( \hat{\ell}_\Im^{(m)}, \hat{s}_\Im^{(m)} \right)$. Therefore, the proposed method performed the search using the following modified Rx vector:

$$\mathbf{y}^{(1)} = \mathbf{y} - \mathbf{h}^{(\hat{\ell}_\Re^{(l)})} s_\Re^{(l)}, l = 1, \cdots, N_\Re. \tag{9}$$

The decision metrics $\mathbf{d}_2$ required in the second part or the Near-ML proposal detector for the MIMO-QSM scheme can be established as follows:

$$\mathbf{d}_2 = \| \mathbf{y}^{(1)} - \mathbf{h}^{(\hat{\ell}_\Im)} \cdot \hat{\mathbf{s}}_\Im \|_F^2, \tag{10}$$

---

**Algorithm 1** QSM real part decoding.

---

**Require:** Channel matrix $\mathbf{H}$, received vector $\mathbf{y}$, $\mathbf{s}_{\Re}$, $n_T$
**Ensure:** The set of tuples ordered $(\hat{\ell}_{\Re}^{(l)}, \hat{s}_{\Re}^{(l)})$
 1: Let $\mathbf{d}_1 = [\cdot]$, $\mathbf{tuple}_{\Re} = [\cdot]$
 2: **for** $i = 1 : n_T$ **do**
 3:     Let $\mathbf{d}_1 = \left[ \mathbf{d}_1 \quad \| \mathbf{y} - \mathbf{h}^{(i)} \cdot \mathbf{s}_{\Re} \|_F^2 \right]$
 4:     **for** $l = 1 : q_{\Re}$ **do**
 5:         Let $\mathbf{tuple}_{\Re} = \left[ \mathbf{tuple}_{\Re} \quad (\hat{\ell}_{\Re}^{(l)} = i, \hat{s}_{\Re}^{(l)}) \right]$
 6:     **end for**
 7: **end for**
 8: Let $[\mathbf{d}_{\Re} \quad \mathbf{ord}_{\Re}] = \mathbf{sort}(\mathbf{d}_1)$ in ascending order.
 9: Order $\mathbf{tuple}_{\Re}$ with the same order of $\mathbf{ord}_{\Re}$
10: Return $\mathbf{tuple}_{\Re}$, $[\mathbf{d}_{\Re} \quad \mathbf{ord}_{\Re}]$

---

For the case of the distance $\mathbf{d}_2$, it was a vector of distances calculated for each valid combinations between the $\hat{\ell}_{\Im}^{\text{th}}$ transmitter antenna $\mathbf{h}^{(\ell_{\Im})}$ and each element of the vector $\mathbf{s}_{\Im}$. This part of the Near-ML algorithm is summarized in Algorithm 2. We denote the $i^{\text{th}}$ column of $\mathbf{H}$ like $\mathbf{h}^{(i)}$ and the $j^{\text{th}}$ row of $\mathbf{h}^{(i)}$ and $\mathbf{y}^{(1)}$ like as $\mathbf{h}^{(i)}(j)$ and $\mathbf{y}^{(1)}(j)$, respectively.

---

**Algorithm 2** QSM imaginary part decoding.

---

**Require:** Channel matrix $\mathbf{H}$, modified received vector $\mathbf{y}^{(1)}$, $\mathbf{s}_{\Im}$, $n_T$, $n_R$ $V_{th2}$
**Ensure:** The optimum tuple $(\hat{\ell}_{\Im}, \hat{s}_{\Im})$, $d_T$
 1: Let $\mathbf{d}_2 = [\cdot]$, $\mathbf{tuple}_{\Im} = [\cdot]$
 2: **for** $i = 1 : n_T$ **do**
 3:     Let $\mathbf{d}_2 = \left[ \mathbf{d}_2 \quad \| \mathbf{y}^{(1)}(1) - \mathbf{h}^{(i)}(1) \cdot \mathbf{s}_{\Im} \|_F^2 \right]$
 4:     **for** $m = 1 : q_{\Im}$ **do**
 5:         Let $\mathbf{tuple}_{\Im} = \left[ \mathbf{tuple}_{\Im} \quad (\hat{\ell}_{\Im}^{(m)} = i, \hat{s}_{\Im}^{(m)}) \right]$
 6:     **end for**
 7: **end for**
 8: Let $[\mathbf{d}_{\Im} \quad \mathbf{ord}_{\Im}] = \mathbf{sort}(\mathbf{d}_2)$ in ascending order.
 9: Order $\mathbf{tuple}_{\Im}$ with the same order of $\mathbf{ord}_{\Im}$
10: Let $lim = q_{\Im} n_T$
11: **for** $i = 2 : n_R$ **do**
12:     Let $d_{min} = \infty$
13:     **for** $m = 1 : lim$ **do**
14:         Let $\hat{\ell}_{\Im} = \ell_{\Im}^{[\mathbf{ord}_{\Im}(m)]}$ and $\hat{s}_{\Im} = s_{\Im}^{[\mathbf{ord}_{\Im}(m)]}$
15:         Let $err = \| \mathbf{y}^{(1)}(i) - \mathbf{h}^{(\ell_{\Im})}(i) \cdot \hat{s}_{\Im} \|^2$
16:         Let $\mathbf{d}_{\Im}(m) = \mathbf{d}_{\Im}(m) + err$
17:         **if** $\mathbf{d}_{\Im}(m) < d_{min}$ **then**
18:             Let $d_{min} = \mathbf{d}_{\Im}(m)$
19:             Let $\hat{\ell}_{\Im} = \ell_{\Im}^{[\mathbf{ord}_{\Im}(m)]}$ and $\hat{s}_{\Im} = s_{\Im}^{[\mathbf{ord}_{\Im}(m)]}$
20:         **else if** $\mathbf{d}_{\Im}(m+1) > V_{th2}$ **then**
21:             $lim = m$
22:             **break**
23:         **end if**
24:     **end for**
25: **end for**
26: Return $[\hat{\ell}_{\Im}, \hat{s}_{\Im}]$, $d_{min}$

---

The complete Near-ML detector is described with detail in Algorithm 3. Each iteration produced a symbol estimation $\left[\hat{\ell}_\Re, \hat{s}_\Re, \hat{\ell}_\Im, \hat{s}_\Im\right]$ with distance $d_T$. Symbol pairs $\left[\mathbf{tuple}_\Re, \hat{\ell}_\Im, \hat{s}_\Im\right]$ whose distance $d_{min}$ were not smaller than the previous ones were skipped. In each iteration, we used the criterion ($V_{th1} = n_R\sqrt{\gamma}$ and $V_{th2} = 2V_{th1}$). The detector used the metrics of the sphere detector to stop the search and discard branches of the tree that were not viable solutions because they exceeded the maximum radius of the detection sphere according to [15,24]. With this modification, the number of branches for each level was adaptive and depended on the SNR and the channel. For these reasons, the proposed algorithm had a significantly reduced complexity.

---

**Algorithm 3** Complete Near-ML detector.

---

**Require:** Channel matrix $\mathbf{H}$, modified received vector $\mathbf{y}$, $\mathbf{s}_\Re$, $\mathbf{s}_\Im$ $n_T$, $n_T$, $q$, $N_b$, $\gamma$, $N_\Re$

**Ensure:** Optimum $\left[\hat{\ell}_\Re, \hat{s}_\Re, \hat{\ell}_\Im, \hat{s}_\Im\right]$

1: Let $d_T = \infty$

2: Let $V_{th1} = n_R\sqrt{\gamma}$ and $V_{th2} = 2V_{th1}$

3: Execute the QSM real part decoding to obtain $\mathbf{tuple}_\Re$, $[\mathbf{d}_\Re \quad \mathbf{ord}_\Re]$

4: Let $lim = length(\mathbf{tuple}_\Re)$

5: **for** $m = 1 : lim$ **do**

6:　　Let $(\hat{\ell}_\Re^{(m)}, \hat{s}_\Re^{(m)}) = \mathbf{tuple}_\Re(m)$

7:　　Let $\mathbf{y}^{(1)} = \mathbf{y} - \mathbf{h}^{\hat{\ell}_\Re}\hat{s}_\Re$

8:　　Execute the QSM imaginary part decoding to obtain $[\hat{\ell}_\Im, \hat{s}_\Im]$, $d_{min}$

9:　　**if** $d_{min} < d_T$ **then**

10:　　　　Let $d_T = d_{min}$

11:　　　　Let $\hat{\ell}_\Re = \ell_\Re^{(m)}$

12:　　　　Let $\hat{s}_\Re = s_\Re^{(m)}$

13:　　　　Let $\hat{\ell}_\Im = \ell_\Im$

14:　　　　Let $\hat{s}_\Im = s_\Im$

15:　　　　**if** $d_{min} < V_{th1}$ **then**

16:　　　　　　**break**

17:　　　　**end if**

18:　　**else if** $d_T > V_{th1}$ and $(V_{th2} < d_{min} < V_{th1})$ **then**

19:　　　　**break**

20:　　**end if**

21: **end for**

22: Return $\left[\hat{\ell}_\Re, \hat{s}_\Re, \hat{\ell}_\Im, \hat{s}_\Im\right]$

---

It is also worth noting that in the proposed algorithm, it is possible to adjust the complexity/BER performance trade-off of the detector the maximum limit of the thresholds $V_{th1}$ and $V_{th2}$. The advantages of our proposal with respect to other similar schemes recently proposed were: It did not require calculating the QR decomposition; therefore, it was less complex. Additionally, it did not require using complex operations; therefore, it was most adequate for hardware implementation.

In the next subsection, BER performance results and the detection complexity of the proposed scheme were compared to the conventional MIMO-QSM scheme for the ML detection algorithm.

Furthermore, the BER performance and the complexity of the proposed low-complexity detection algorithm were analyzed.

### 3.1. BER Performance Comparison of the MIMO-QSM Scheme

In this subsection, two different configurations were used in order to compare the BER performance of the proposed MIMO-QSM scheme for the ML detection under uncorrelated Rayleigh fading channels. The systems were analyzed considering the same spectral efficiency, the same number of Tx and Rx antennas, and a normalized transmission power in the transmitter. For all computer simulations, we targeted a BER of $10^{-4}$.

Figure 2 shows the performance comparison for the optimal ML and the proposed low complexity Near-ML algorithm for the MIMO-QSM scheme using the $2 \times 2$ and $8 \times 8$ configuration with QPSK modulator. For both cases, the proposed detector performed very near to the ML algorithm, and for this reason, we called our proposal detector Near-ML.

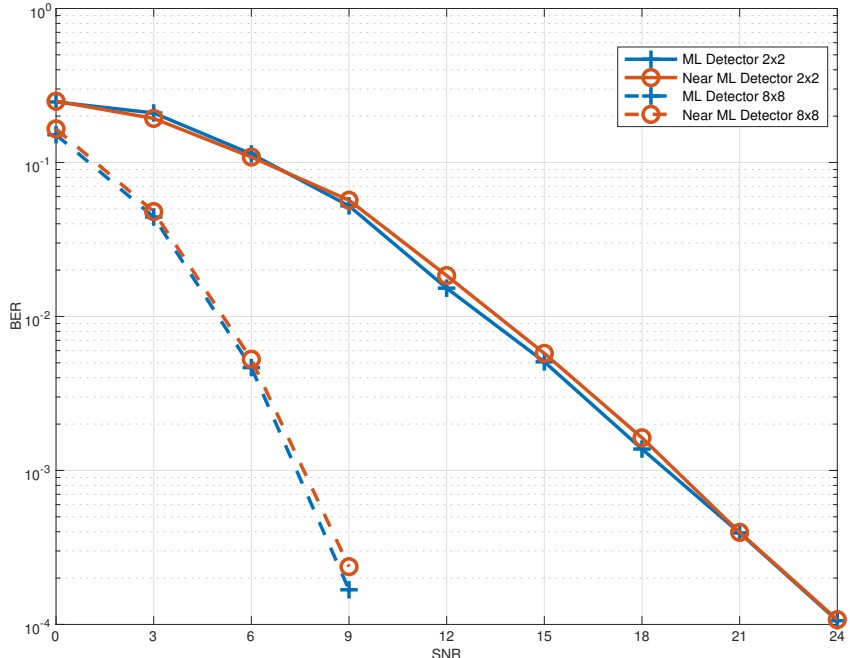

**Figure 2.** Performance comparison for an $2 \times 2$ and $8 \times 8$ configuration at $m_{QSM} = 4$ and $m_{QSM} = 8$ bpcu respectively.

### 3.2. Complexity

The ML detection complexity $\eta$ for the ML criterion in (6) was measured in terms of complex operations (CO). One arithmetic operation was considered as one CO; also, one comparison was considered as one CO. The lattice in the system had $2^{m_{QSM}}$ points. Subtraction, obtaining the square, and finding the minimum in (6) resulted in $2^{m_{QSM}+2}$ CO. Considering $n_R$ receive antennas, the complexity for ML in (6) can be approximated by:

$$\eta \approx 2^{m_{QSM}+2} n_R. \tag{11}$$

Table 2 shows a comparison of the complexity for two MIMO configurations. In Table 2, the QSM scheme with ML detector is considered as the reference with 100% of complexity for an SNR of 9 dB. The last column shows the complexity of the proposed Near-ML algorithm where it is observed how this detector outperformed significantly the QSM ML detector in terms of complexity.

**Table 2.** Comparison of complexity ($\eta$).

| Scheme/$\eta$ | QSM ML | QSM Near ML |
|---|---|---|
| 4 bpcu $2 \times 2$ QPSK | 256 (100%) | 282 (73%) |
| 8 bpcu $8 \times 8$ QPSK | 12,288 (100%) | 1475 (12%) |

The results showed that the proposed Near-ML algorithm performed very near to the optimal one with the advantage of a reduction in detection complexity of 88%.

## 4. Proposed Hardware Architecture for the Digital QSM Detector

A hardware architecture that implements the detection algorithm presented in Section 3 was proposed. In order to illustrate it, the top module is shown first, and then, the internals of each module are explored.

The design of the architecture followed a top-down approach, and it was composed of the data-path that included four modules and a control unit. The detector is presented in Figure 3. The **DET1** and **DET2** modules implemented the first and second parts of the detection process, described in Algorithms 1 and 2, respectively; the **SORT** module performed the sort operation used in the three algorithms; the **FD** module compared the distance metrics needed to determine the received symbol (inside the *for* cycle in Algorithm 3); and the **CTRL** module provided the timing signals and activation flags for the data-path.

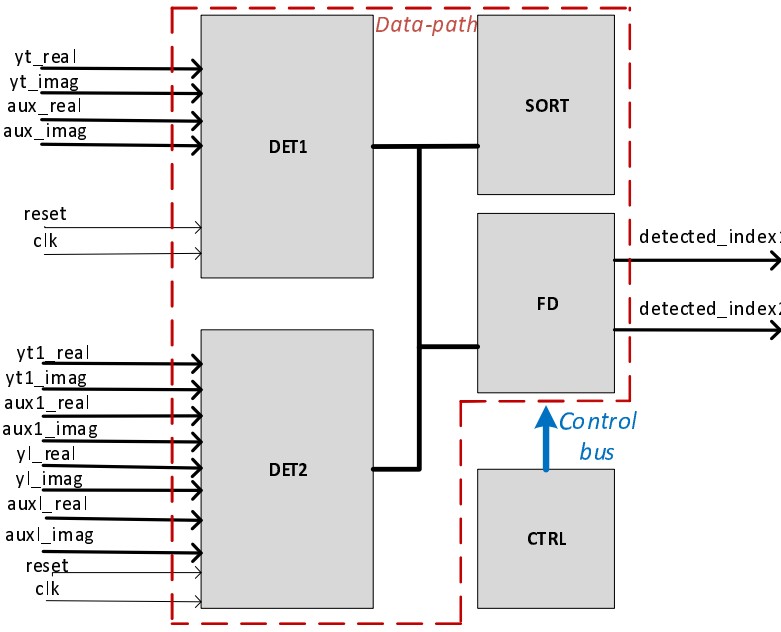

**Figure 3.** QSM detector architecture.

From this point onwards, a bold line represents a bit vector, whereas a thin line represents a single bit.

### 4.1. Top View of the QSM Detector Architecture

Using the description of the QSM communication system in Section 2.1 and the architecture proposed in Figure 3, a top view of the detector is presented considering the following parameters:

- *NT*: number of Tx antennas.
- *NR*: number of Rx antennas.
- *WL*: word length used for fixed point representation.
- *IP*: amount of bits used to represent the integer part of data.
- *FP*: amount of bits used to represent the fractional part of data. Result of subtracting *WL* and *IP*.
- *HXQC*: number of columns of an auxiliary matrix needed for calculations in **DET1**.
- *HXQ1S*: size of an auxiliary vector needed for calculations in **DET2**.

Considering Table 3, which summarizes the inputs and outputs of the whole system, the detection process started when the received signal *y* was mapped into ports *yt_real* and *yt_imag*. This input signal was then processed by **DET1**, which executed Algorithm 1 in hardware. After that, **DET2** received data in its *yt1_real* and *yt1_imag* (equivalent to the modified received vector **y**) ports to continue with Algorithm 2.

**Table 3.** Inputs and outputs of the detector.

| Input | Size (Bits) | Goes into | Description |
|---|---|---|---|
| yt_real | $WL \times NR$ | **DET1** | Real part of the received data, **y**, in antennas for **DET1** |
| yt_imag | $WL \times NR$ | **DET1** | Imaginary part of the received data, **y**, in antennas for **DET1** |
| aux_real | $WL \times NR \times NT$ | **DET1** | Real part of the operation $\mathbf{h}^{(i)}\mathbf{s}$ for **DET1** (Algorithm 1, Line 3) |
| aux_imag | $WL \times NR \times NT$ | **DET1** | Imaginary part of the operation $\mathbf{h}^{(i)}\mathbf{s}$ for **DET1** (Algorithm 1, Line 3) |
| yt1_real | $WL$ | **DET2** | Real part of the received data, **y**, in antennas for **DET2** |
| yt1_imag | $WL$ | **DET2** | Imaginary part of the received data, **y**, in antennas for **DET2** |
| aux1_real | $WL \times HXQ1S$ | **DET2** | Real part of the operation $\mathbf{h}^{(i)}\mathbf{s}$ for **DET2** (Algorithm 2, Line 3) |
| aux1_imag | $WL \times HXQ1S$ | **DET2** | Imaginary part of the operation $\mathbf{h}^{(i)}\mathbf{s}$ for **DET2** (Algorithm 2, Line 3) |
| yl_real | $WL$ | **DET2** | Real part of the received data, **y**, in antennas for **DET2_P2** |
| yl_imag | $WL$ | **DET2** | Imaginary part of the received data, **y**, in antennas for **DET2_P2** |
| auxl_real | $WL$ | **DET2** | Real part of the operation $\mathbf{h}^{(i)}\mathbf{s}$ for **DET2_P2** (Algorithm 2, Line 14) |
| auxl_imag | $WL$ | **DET2** | Imaginary part of the operation $\mathbf{h}^{(i)}\mathbf{s}$ for **DET2_P2** (Algorithm 2, Line 14) |
| **Output** | **Size (Bits)** | **Goes to** | **Description** |
| detected_index1 | $log_2(NT \times HXQC)$ | Detector output | First detected index |
| detected_index2 | $log_2(NT \times HXQC)$ | Detector output | Second detected index |

The sort operation was utilized by both detection processes; therefore, **SORT** can be used by **DET1** and **DET2**.

At the final step of the detection process, the **FD** module compared the current detection results with predefined parameters ($V_{th1}$, $V_{th2}$, $d_{min}$, in Algorithm 3); if the process matched these parameters, the current results became the final results; otherwise, the detection process was restarted until the conditions were met. The **CTRL** module controlled and synchronized the whole interoperability of the architecture.

The outputs *detected_index1* and *detected_index2* represent the pair *[transmit antenna, symbol sent]* for QSM.

In what follows, the modules of the detector are described in detail.

### 4.2. DET1 *Module*

The detection started with DET1, implementing Algorithm 1. Its architecture is shown in Figure 4, and an explanation of the inputs and outputs is in Table 4.

Modules:

- Norm (NORM): operation of Euclidean distance.
- RegisterArray (RA): register array for the results of NORM.
- SortingRegs (SREGS): register array for the results of SORT.

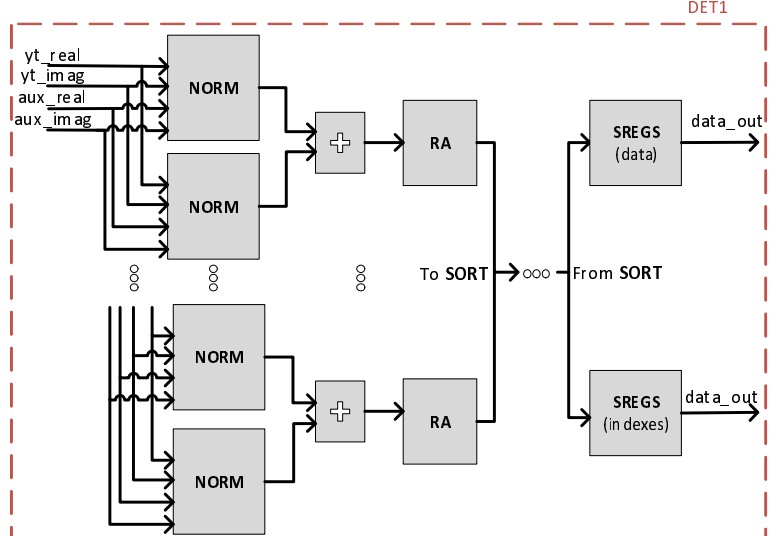

**Figure 4.** Architecture of DET1.

**Table 4.** Inputs and outputs of DET1.

| Input | Size (Bits) | Comes from | Description |
|---|---|---|---|
| yt_real | $WL \times NR$ | System input | Real part of the received data, $\mathbf{y}$, in antennas |
| yt_imag | $WL \times NR$ | System input | Imaginary part of the received data, $\mathbf{y}$, in antennas |
| aux_real | $WL \times NR \times NT$ | System input | Real part of the operation $\mathbf{h}^{(i)}\mathbf{s}$ for DET1 (Algorithm 1, Line 3) |
| aux_imag | $WL \times NR \times NT$ | System input | Imaginary part of the operation $\mathbf{h}^{(i)}\mathbf{s}$ for DET1 (Algorithm 1, Line 3) |
| **Output** | **Size (Bits)** | **Goes to** | **Description** |
| data_out (data) | $WL$ | FD | Final data of DET1 ($\mathbf{d}_{\Re}$ in Algorithm 1) |
| data_out (indexes) | $log_2(NT \times HXQC)$ | FD | Final index of DET1 ($\mathbf{ord}_{\Re}$ in Algorithm 1) |

When the detection started, the NORM modules computed the Euclidean distance operation between (*yt_real*, *yt_imag*) and (*aux_real*, *aux_imag*) corresponding to the norm operation presented in Line 3 of Algorithm 1.

The results were then added and stored in the corresponding RA registers in an orderly manner. When every register had data stored, these results were sent to the SORT module in a parallel way, as better described in Section 4.4.

When the sorted data returned, as seen in "From SORT" in Figure 4, the data and indexes were stored, in the same order as they came out of the sorting module, in their respective SREGS, which was another register array.

*4.3. **DET2** Module*

The second detection process, **DET2**, was at the same time divided into two parts representing the *for* cycles in Lines 2 and 11, respectively, of Algorithm 2. Figure 5 shows the first part (**DET2_P1**) with the first **NORM** blocks and the second part (**DET2_P2**) with the last block. When **DET2_P1** ended, **DET2_P2** immediately started. Table 5 shows the inputs and outputs of the module.

Modules:

- Norm (**NORM**): operation of Euclidean distance.
- RegisterArray for **DET2** (**RA_DET2**): register array for the results of **NORM** in **DET2_P1**.
- SortingRegs_SP (**SREGS_SP**): register array for the sorting results in **DET2**.
- SortingRegs_SP2 **SREGS_SP2**: register array for the sorting indexes results in **DET2**.
- RearrangerP(**RP**): rearranger for the sorting indexes.

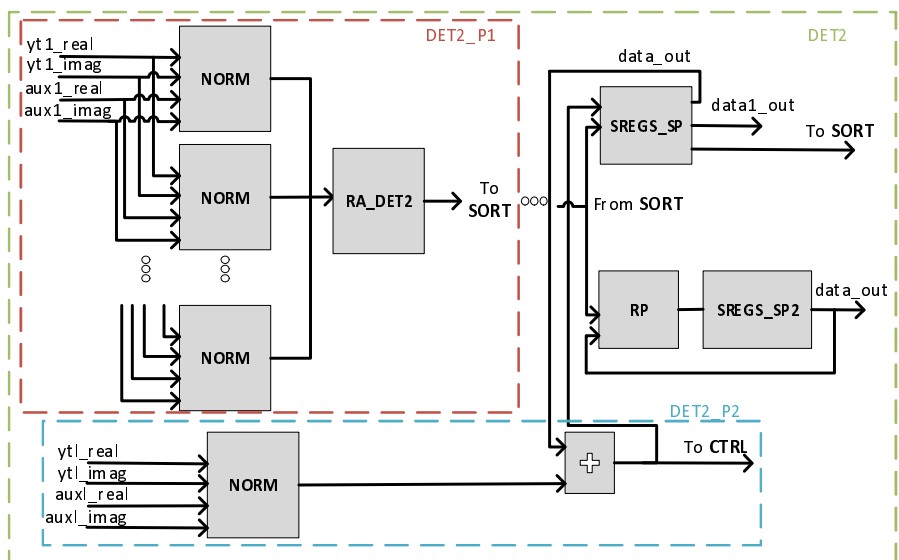

**Figure 5.** Architecture of **DET2**.

After the calculations with the **NORM** modules were done, **RA_DET2** sent its stored data to **SORT**. When the data came back from the module, they were split into the data vector and index vector to be stored in **SREGS_SP** and **SREGS_SP2**, respectively.

**SREGS_SP** was a register array that had both serial and parallel inputs and outputs and stored the sorting data. It also had outputs like *data1_out* that fed the control module, equivalent to $\mathbf{d}_\Im(m+1)$ in Algorithm 2, Line 19.

**SREGS_SP2** stored the indexes given by the **SORT** module. It only had a parallel input, but serial and parallel outputs; this configuration helps when you need to reorder all indexes (parallel output, as in Line 9 of Algorithm 2) or you need to read only one index.

In **DET2_P2**, another **NORM** module was used, and its result was added to one of the already stored in **SREGS_SP**, then fed back to the same register and sent to sorting again, emulating the behavior of Lines 14 and 15 of Algorithm 2.

When coming back from sorting, **RP** rearranged the old indexes stored in **SREGS_SP2** based on the new ones and replaced them. For example, if the stored vector was $[3, 1, 2, 4]$ and the new indexes were $[2, 1, 4, 3]$, then the rearranged vector would be $[1, 3, 4, 2]$.

**Table 5.** Inputs and outputs of DET2.

| Input | Size (Bits) | Comes from | Description |
|---|---|---|---|
| yt1_real | *WL* | System input | Real part of the received data in antennas minus the influence of the detected data in DET1 (as in Line 7 of Algorithm 2) |
| yt1_imag | *WL* | System input | Imag.part of the received data in antennas minus the influence of the detected data in DET1 (as in Line 7 of Algorithm 2) |
| aux1_real | $WL \times HXQ1S$ | System input | Real part of the operation $\mathbf{h}^{(i)}\mathbf{s}$ for DET2 (Algorithm 2, Line 3) |
| aux1_imag | $WL \times HXQ1S$ | System input | Imaginary part of the operation $\mathbf{h}^{(i)}\mathbf{s}$ for DET2 (Algorithm 2, Line 3) |
| yl_real | *WL* | System input | Real part of the received data in the remaining antennas for DET2_P2 |
| yl_imag | *WL* | System input | Imaginary part of the received data in the remaining antennas for DET2_P2 |
| auxl_real | *WL* | System input | Real part of the operation $\mathbf{h}^{(i)}\mathbf{s}$ for DET2_P2 (Algorithm 2, Line 14) |
| auxl_imag | *WL* | System input | Imaginary part of the operation $\mathbf{h}^{(i)}\mathbf{s}$ for DET2_P2 (Algorithm 2, Line 14) |
| **Output** | **Size (Bits)** | **Goes to** | **Description** |
| data_out (data) | *WL* | FD | Final data of DET2 ($\mathbf{d}_\Im$ in Algorithm 2) |
| data_out (indexes) | $log_2(NT \times HXQC)$ | FD | Final index of DET2 ($\mathbf{ord}_\Im$ in Algorithm 2) |

### 4.4. SORT *Module*

The SORT module was the most used during the detection process. Sorting networks were chosen as the option for sorting in FPGA [25].

A sorting network is one of the most efficient and traditional ways of sorting in FPGA. They are attractive due two main reasons: they do not require control instructions, and they are relatively easy to parallelize due to the simplicity of the data flow. Sorting networks are adequate for sorting short arrays, the length of which is known beforehand.

According to the detection algorithm, besides sorting, the network must indicate in which position of the array the elements were originally, before entering the network and being sorted; similar to how MATLAB does it with its integrated function $[B, I] = sort(A)$ [26]. There was already a work that addressed this problem in hardware [27], so the architecture proposed there was used here.

The implemented sorting network consisted of purely combinational comparators so, as the network grew according to the sorting needs, the critical route of the system increased as well.

Figure 6 shows the structure of the sorting network, and Table 6 describes the inputs and outputs of the module.

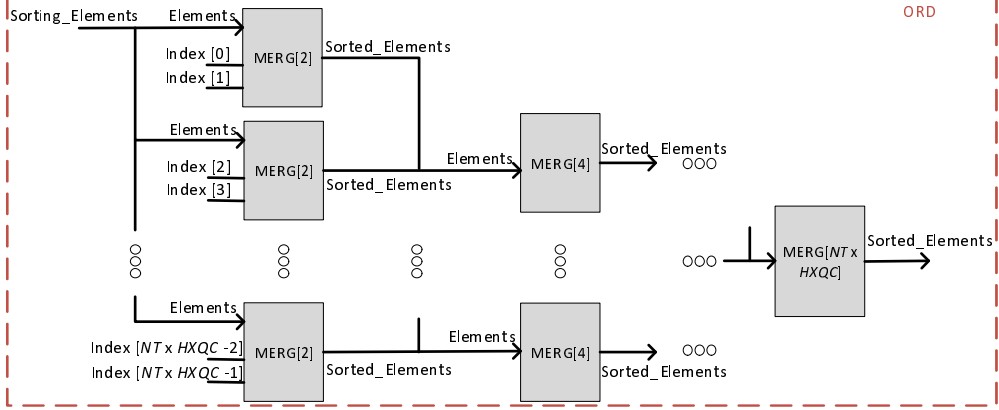

**Figure 6.** Architecture of SORT.

**Table 6.** Inputs and outputs of SORT.

| Input | Size (Bits) |
|---|---|
| Sorting_Elements | $WL \times NT \times HXQC$ |
| Elements | $WL$ |
| Index | $log_2(NT \times HXQC)$ |
| **Output** | **Size (Bits)** |
| Sorted_Elements | $WL + log_2(WL \times HXQC) \times E2S \times NT \times HXQC$ |

The vector Sorting_Elements came from the registers in DET1 and DET2, and Sorted_Elements was composed of the sorted data and their respective indexes that were going to be stored in SREGS.

### 4.5. FD *and* CTRL *Modules*

The FD module made the decision of accepting the current detected indexes as valid or not. It achieved this by comparing the detection results with predefined parameters specific to QSM detectors ($V_{th1}$, $V_{th2}$, $d_{min}$, in Algorithm 3). In case the results were accepted, the detection process ended; otherwise, DET2 started over with different data. This was implemented with counters and a state machine, so it emulated the behavior of the *for* cycle in Algorithm 3.

The CTRL module grouped three independent modules that controlled the three main parts of the detector (DET1, DET2_P1, and DET2_P2).

The control modules for DET1 and DET2_P1 were composed mainly of counters that checked the number of elapsed clock cycles, representing their respective *for* cycles in the algorithms.

The control module for DET2_P2 differed from the other two control modules. This one consisted of a state machine and counters that performed the *for* cycles in Algorithm 2, specifically Lines 11 and 13. As is seen in Line 13, the *lim* variable was known until execution time and changed depending on the data, the reason why a state machine was required for proper control.

## 5. Analysis of Hardware Implementation and Verification Results

### 5.1. Hardware Budget

In order to implement the proposed design, an Intel-Altera Cyclone IV EP4CE115 FPGA was used. In Table 7, the amount of resources, maximum frequency, and throughput, are shown for two representative cases of the QSM communication systems. The number in brackets represents the percentage of resources used out of the total available in the FPGA device.

Table 8 presents a breakdown of post-synthesis resources used by the different modules of the architecture for the 2 × 2 QPSK and 2 ×2 16-QAM configurations, respectively. Naturally, the amount of resources went up as the order of modulation and the number of antennas increased.

According to the results, the SORT block was the module with the greatest amount of hardware resources. It used only combinational components (as it was based on sorting networks) [27] to make a comparison between elements and increased in size as the number of elements to sort rose. Given these characteristics, the critical route of the whole implementation was established by this module and affected the general performance of the architecture.

**Table 7.** Overall implementation results of the detector in a Cyclone IV FPGA.

| Configuration | Logic Elements | Embedded Multipliers | Max Frequency | Throughput |
|---|---|---|---|---|
| 2 × 2, QPSK | 2385 (2%) | 28 (5%) | 37.11 MHz | 416,966 ops |
| 2 × 2, 16-QAM | 6863 (5%) | 36 (6%) | 20.74 MHz | 171,404 ops |

Due to the algorithm being inherently recursive, the number of clock cycles was data dependent. Table 7 shows the max frequency considering the slow 1200 mV 0°C model; and throughput, which is the number of detection processes that can be done per second in the worst case.

**Table 8.** Resources used per module of the architecture.

| 2 × 2, QPSK Detector Module | LCCombinational | LC Registers | DSPElements |
|---|---|---|---|
| DET1 | 308 | 152 | 16 |
| DET2 | 408 | 140 | 12 |
| SORT | 1279 | 0 | 0 |
| FD | 33 | 32 | 0 |
| CTRL | 69 | 37 | 0 |
| **2 × 2, 16-QAM Detector Module** | **LC Combinational** | **LC Registers** | **DSP Elements** |
| DET1 | 386 | 320 | 16 |
| DET2 | 835 | 288 | 20 |
| SORT | 4696 | 0 | 0 |
| FD | 34 | 36 | 0 |
| CTRL | 84 | 42 | 0 |

*5.2. Simulation Results*

In order to verify the results, a MATLAB implementation was used to generate random input vectors written in text files for the architecture. These files were fed into a test bench that controlled the reading of the test vectors and the writing of the results.

Figure 7 summarizes the testing process to obtain the results and compare them with the outputs of the MATLAB algorithm.

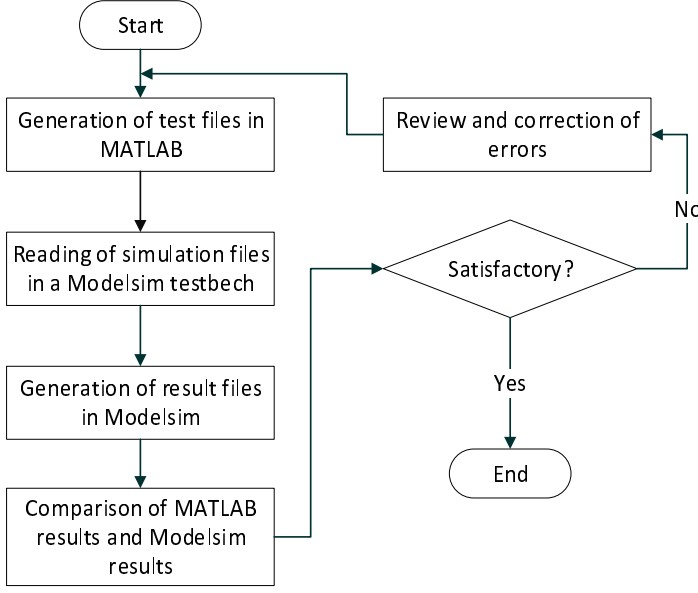

**Figure 7.** Process of the generation of simulation results.

Fixed point simulations were performed in order to determine the ideal *IP* and *FP* parameters depending on the BER performance of the algorithm. Figure 8 shows the comparison for different configurations of word length against the floating point model used (4 × 4, QPSK). Taking as a reference Figure 8, the fixed point format *IP* = 5 and *FP* = 11 had a close performance to the floating point model.

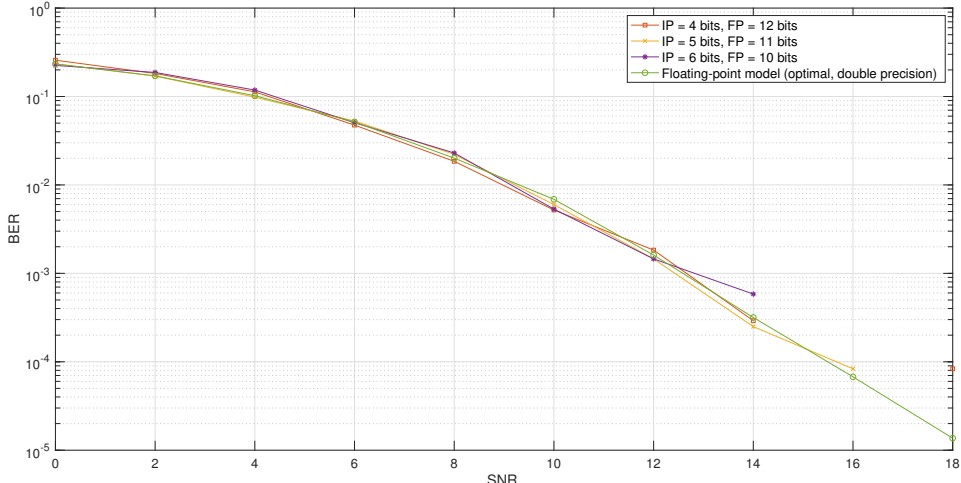

**Figure 8.** Fixed point analysis for the algorithm, 16-bit word length, 4 × 4, QPSK.

Figure 9 shows the timing diagram of the detection process. When start = 1, *yt_real* and *yt_imag* were processed by **DET1** and after five cycles in the case of a QPSK modulation, the flag *done_detection1* was set to one (*END_DET1* in Figure 9), indicating that **DET2_P1** could start. At this point, Algorithm 1 was performed.

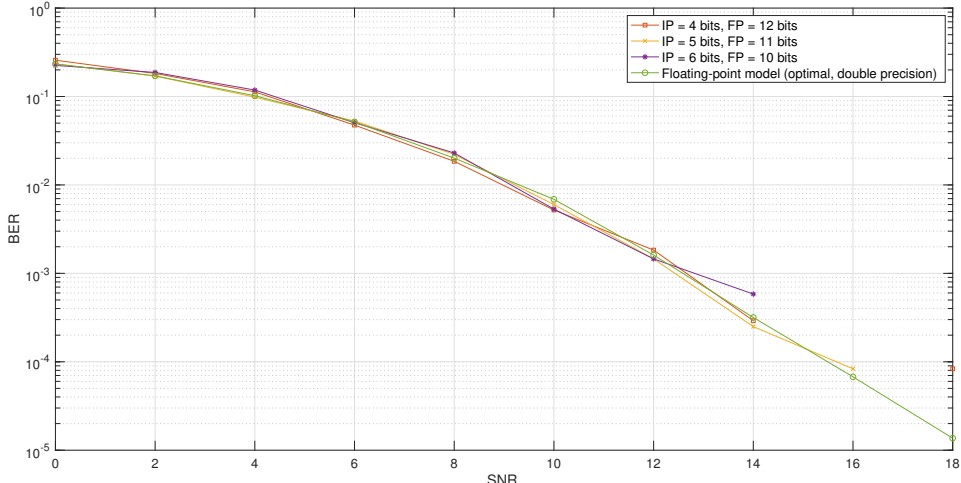

**Figure 9.** Simulation in Modelsim of the proposed detector.

The **DET2_P1** module read the *yt1_real* and *yt1_imag* inputs and processed them according to Algorithm 2. Immediately after, **DET2_P2** did the same with the *yl_real* and *yl_imag* inputs and set *done_detection2* to one (*END DET2*) when it finished, informing the **FD** module that the data were ready for it to make a decision. At this point, Algorithm 2 was performed.

The **FD** module required one clock cycle to make the decision of accepting the current detected indexes or not. In the case of this simulation, said indexes were not accepted at first, so it sent the appropriate signals to start **DET2** over.

**DET2** read its respective signals and was performed again. When it finished (represented by the rightmost *END DET2* in Figure 9), the **FD** module took another clock cycle to decide. This time, the detected indexes were accepted, as the *finished_detection* flag was raised, meaning that the detection process (Algorithm 3) was over and the detected data (*FINAL DATA* in Figure 9) were valid.

## 6. Conclusions

A low complexity detection algorithm based on a tree search and spherical detection, in the context of MIMO QSM transmission, was presented. It was shown that the proposed algorithm achieved a similar performance to the ML detector, but with a significant complexity reduction in terms of the operations required in its software and hardware implementation. Fixed point analysis showed that BER performance was maintained on the detection process, allowing a simpler hardware implementation rather than the hardware needed for an implementation using floating point precision. The novel hardware architecture showed the feasibility of the hardware implementation of the proposed algorithm using fixed point precision and the process of transformation from the algorithm to hardware architecture.

A possible workaround for the critical route would be the implementation of pipeline stages inside the sorting module in order to improve the maximum frequency and throughput significantly, even if the added registers would mean a redesign of the control unit.

**Author Contributions:** Conceptualization, I.L., J.C. and L.P.-E.; Methodology, I.L., J.C. and L.P.-E.; Software, I.L. and J.C.; Experimentation, I.L., J.C., L.P.-E. and O.L.-G.; Validation, I.L., J.C., L.P.-E. and O.L.-G.; Formal Analysis, I.L., J.C., L.P.-E., O.L.-G. and A.G.; Investigation, I.L., J.C., L.P.-E., O.L.-G. and A.G.; Resources, J.C. and A.G.; Data Curation, I.L.; Writing—Original Draft Preparation, I.L.; Writing—Review & Editing, I.L., J.C., L.P.-E. and O.L.-G.; Visualization, I.L., J.C., L.P.-E., O.L.-G. and A.G.; Supervision, I.L., J.C., L.P.-E. and O.L.-G.; Project Administration, J.C. and A.G.; Funding Acquisition, J.C. and A.G.

**Funding:** The present article was jointly funded by PFCE 2019, CONACYT scholarship and PROFAPI 2019.

**Conflicts of Interest:** On behalf of all authors, the corresponding author states that there is no conflict of interest.

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
