# Peer review of "Fast Scalable Architecture of a Near-ML Detector for a MIMO-QSM Receiver"

_electronics, doi:10.3390/electronics8121509_

Round 1
Reviewer 1 Report
The work undertaken is done well and presented in an adequate manner. However, the structure of the paper can be significantly improved, where the clarity of work and stages undertaken and presented would improve by doing so. The authors should include a methodology section and further highlight the contributions and impact of work presented.
Currently, the article is unclear on the work being undertaken. The authors present a limited and short initial literature review.
The article describes MIMO-QSM receiver structures proceeding to an 'optimal' MIMO-QSM architecture. Then a generic architecture is described in terms of performance (BER) with hardware implementation.
The implementation methodology or investigation undertaken is unclear, until you reach section 5 (page 14 of 17). This is the first detailed instance describing the implementation or simulation of graphs received earlier in the article. At this point, it is assumed the previous models were implemented within matlab as it is not stated.
The hardware implementation is extremely limited (~2 pages, where ~1 page is tables and figures). The theoretical hardware design premises (in section 4) are presented well, but should be linked towards the hardware implementation (FPGA design).
The FPGA results presented are significantly limited and naive. A modelsim simulation is presented with limited discussion. Similarly, authors present the Cyclone IV results with limited discussion - there is no details whether this is estimated or post-synthesis, leaving readers to assume.
The authors do not put the results achieved into context, nor explore any significant investigation of implementation of note. Currently, the level of investigation of hardware would be best suited to a conference article summarising an implementation.
The authors should expand the simulation waveform (latency, throughput, ...) and hardware resources (LUTs, power, ...) for the various high-level analysis of algorithm fixed-point length, as presented in Figure 8.
Furthermore, as the architetcure is presented as being 'optimal'. The paper fails to determine and verify the 'optimal' architecture achieved with only QPSK and 16-PSK investigated and no other architecture or word lengths investigated to validate the 'optimal' status.
Author Response
Plese see the attachment
Best regards
Joaquin Cortez

Reviewer 2 Report
Please refer to the attached file.

Author Response
Please see the attachment
Best regards
Joaquin Cortez

Round 2
Reviewer 1 Report
The authors have addressed all concerns and issues raised to an adequate standard for publication. The authors may wish to consider a detailed hardware analysis/comparison for word length with various implementations as their future work.